# Sensory and Conceptual Aspects of Ingredients of Sustainable Sources—Finnish Consumers’ Opinion

**DOI:** 10.3390/foods9111669

**Published:** 2020-11-15

**Authors:** Saara Lundén, Anu Hopia, Laura Forsman, Mari Sandell

**Affiliations:** 1Functional Foods Forum, University of Turku, FI-20014 Turku, Finland; anuhop@utu.fi (A.H.); laura.forsman@utu.fi (L.F.); mari.sandell@utu.fi (M.S.); 2Department of Food and Nutrition, University of Helsinki, FI-00014 Helsinki, Finland

**Keywords:** consumer attitudes, ingredients, pleasantness, sensory, sustainability

## Abstract

Sustainable strategies that enable development of alternative sustainable novel ingredients for food are needed to ensure adequate resources for food in the future. Determining consumer attitudes and acceptance of novel ingredients is essential for wider usage of products including these ingredients. The purpose of the study was to reveal consumers’ perspectives on novel, and partly traditional but marginally utilized, ingredients to be used in regular cooking and their sensory characteristics and nutritional and environmental aspects. Consumer attitudes were obtained with two online consumer surveys. Consumer surveys revealed the most interesting ingredients. Plant-based ingredients are preferred over raw materials of animal or insect origin and these are also perceived as more pleasant. Plants were also regarded as credible, ecological, natural, healthy and nutrient-rich. Finnish consumers are not ready to adopt insects into their diet. Neither synthetic meat nor three-dimensional printed food have potential without further knowledge or experience of consumers. Findings of this research give baseline information on consumer attitudes towards novel ingredients. Further research is needed to investigate the perceived pleasantness when the potential ingredients are tasted.

## 1. Introduction

A crisis of resource sustainability is facing us as a population. A key driver for this is the socio-culturally defined selectivity in consumption habits within developed countries [1]. Sustainable strategies that enable increasing agricultural production and the development of alternative sustainable novel ingredients for food are needed [2]. Determining consumer attitudes and acceptance of novel ingredients of sustainable sources is essential for the commercialization and wider usage of products including these ingredients.

Environmental preservation has become one of the main concerns of consumers [3]. Consumers’ interest in environmentally friendly products has grown over recent decades [3,4,5,6,7]. However, regarding food choices, health-related issues and food origin are more preferred reasons than environmental awareness [8]. Other major dichotomies consumers use in categorizing food ingredients are natural vs. unnatural and positive vs. negative [9] which might hinder the acceptance of many underutilized ingredients. Furthermore, consumers perceive ingredients more risky when they are not familiar with them [9,10]. Thus, though consumers might pursue more sustainable food choices, many potential new ingredients suffer from unfamiliarity and various bad images, which makes their application in new products risky for the food producers. Therefore, it is essential to gather understanding about the preliminary images consumers have about new sustainable ingredients and the differences between these images, in order to opt for the most potentially acceptable ingredients for the development of new sustainable food products.

Insects have been allowed to be used as food in the European Union (EU) since the beginning of 2018 after the Novel Food Regulation (Regulation (EU) 2015/2283) came into force regarding edible insects [11]. Opinions and acceptance regarding insects as food in Western countries have recently gained a lot of academic interest (see e.g., [12,13,14,15,16,17,18]). Insects are eaten in numerous countries around the world and insect-based food was estimated to include 2000 species of edible insects [19]. However, Western consumers, such as the Europeans, are only beginning to familiarize themselves with insects as food [14,15]. Rejection of insects as food is mainly caused by disgust which is primarily not based on sensory properties of insects but on knowledge of the history and nature of a potential food [14]. This is supported by the results of Megido et al. [12] who reported high willingness to eat and cook insects as food in the near future after tasting insect preparations. Other predictors for the acceptance of edible insects are previous insect consumption, food neophobia, gender, sensation seeking and food technology neophobia [20]. Willingness to consume insects is found to be culturally relative and differing even in European subcultures [15,21]. Consumers in Northern Europe have a more positive attitude towards insect food compared to consumers in Central Europe [15].

Food wastes and by-products are another possible group of novel sustainable ingredients. A circular economy model can be implemented to the food sector by recycling its by-products and hence creating added value with fewer resources [22]. By-products have been considered as low value and discarded without further processing [23]. Recently, food by-products have been studied as a source of sustainable ingredients or bioactive compounds to be used in functional foods as they can have high nutritional value [23,24].

Three-dimensional food printing has suggested to have implications for future food development. Implications include reducing food waste using food that is usually discarded, such as fruits and vegetables having poor quality in appearance [25]. However, little research has been conducted on consumer perception of three-dimensional-printed (3D-printed) food. Manstan and McSweeney [25] reported a positive attitude towards 3D-printed food, even when compared to a conventional counterpart. Three-dimensional printed food products were believed to be healthier and less processed than conventional food products. Results by Brunner et al. [16] oppose this finding as they found Swiss participants to have a negative overall attitude towards 3D-printed food.

Wild food plants have traditionally been used around Europe, but there has been a dramatic loss of traditional knowledge and practices and the use of these plants in nutrition is very low [26]. However, usage is highly dependent on the region and culinary culture [27,28]. In Mediterranean culinary culture, wild plants are still often used as a part of diet [28]. Food made of cultivated plants and bought from the supermarket appears on the table with relatively little effort, while collecting wild species is more time consuming and season-dependent, thus making them less convenient to be used in everyday cooking [26]. There is a live tradition to use wild berries and mushrooms in Sweden and Finland as these are freely available resources for everyone thanks to legal right of access to private land [26,29]. Wild berries are also used in the food industry and restaurants. Wild plants have recently been promoted by avant-garde restaurants in Northern Europe [26].

When novel ingredients are introduced to consumers, they can potentially cause neophobia, that is fear and refusal of new food [30]. Neophobia limits individuals’ readiness to try new foods and thus restricts the marketability of new ingredients [31]. It is possible that neophobia explains the common thread among all these novel foods or ingredients.

The purpose of the study was to reveal consumers’ perspectives on novel food sources for discovering new potential raw materials for food products and cooking. Our hypothesis was that consumers differ in their opinions on plant-based and other ingredients. The key motivator for this research was to find out the potential barriers and drivers for these food ingredients. Our context of the research was in attitudes of consumers towards novel ingredients and willingness to try and adopt them in cooking and food products. The ingredients need to be either novel or traditional but marginally utilized. Another important aspect to consider in relation to acceptance of novel food ingredients is consumers’ motivation to eat them; such aspects were measured by asking a number of questions related to sensory characteristics and to nutritional and environmental aspects. Furthermore, differences between consumer groups were investigated to achieve a more extensive understanding of the attitudes and to identify possible groups of early adopters of the ingredients.

## 2. Materials and Methods

### 2.1. Preliminary Stage

The basis of the study was the involvement of consumers in every stage of the process. In the preliminary stage the consumers were engaged in collecting ideas for novel ingredients. This was implemented in a local food fair in autumn 2017 (total number of visitors approximately 20,000). Visitors of the annual food fair were encouraged to write down their ideas on the topic “What are we going to eat from nature 2027?”. The list of raw materials (around 100) collected from consumers was supplemented by authors with ideas based on literature and insights from media to increase variation and include some current raw materials. A total of 81 raw materials, presented in Figure 1, were included in the following consumer survey.

### 2.2. Consumer Survey 1

A consumer survey was applied to discover the interest of Finnish consumers towards novel raw materials as food ingredients. The consumer survey was distributed as an online survey with Compusense Cloud (Compusense Inc., Guelph, ON, Canada). Randomly, 30 of the 81 raw materials were presented for each consumer in randomized order. Consumers were asked to choose at least five of the presented raw materials that they would be interested in using in cooking or eat. There was no maximum limit for choices. Gender and age were collected as demographic information of the participants.

Adult volunteers participating in the survey were recruited from a consumer register administered by the University of Turku. There was no exclusion or inclusion criteria for participation. Participants replied to the survey anonymously and they were not rewarded with any incentive. The study was conducted following the ethical principles of the Declaration of Helsinki. Consumer survey 1 worked as a pretest for consumer survey 2.

### 2.3. Consumer Survey 2

Another consumer survey was implemented in December 2017 to further investigate the opinion of Finnish consumers on these novel ingredients. Based on the results of the first consumer survey the ingredients were chosen for the second survey. Ten raw materials were included in the survey. The list of the 81 raw materials in the first survey also included raw materials which are already regularly in use in Finnish diets. Therefore, seeds and pulse were excluded from the next survey even though they were the most interesting raw materials according to the results. The following five most interesting raw materials were included in the survey: nettle, berry bush leaves, spruce or pine shoots, leaves and stem of broccoli and cauliflower and clover. Root vegetable tops were combined with broccoli and cauliflower parts to broaden the selection of the raw materials. These represent the generally wasted parts of the vegetables commonly used in Finnish food culture. This combination excluded peels which were considered less interesting.

Plant-based raw materials were abundant on the original list of the 81 raw materials. These were also selected as the most interesting ones. Nevertheless, the most interesting raw materials of animal or insect origin were considered justified to be included to the survey to obtain a broader impression on consumers’ opinions on the subject. The plant-based novel ingredients considered most interesting based on consumer survey 1 did not include significant protein sources. This was another argument to include raw materials of animal and insect origin. Whey protein and milk protein were found to be the most interesting raw materials of animal origin. These were excluded from the survey based on their prevalence in food manufacturing at present. Crickets, beeswax and ants were the most interesting insect-based raw materials, and they were included in consumer survey 2. Synthetic meat and eggshells were the most interesting raw materials of animal origin. Three-dimensional-printed food was included to the list of raw materials of the second consumer survey. This was argued for, though 3D-printed food is not a raw material, but is a novel method for utilizing both plant-based and animal-based raw materials, as well as for processing by-products into edible food in the future. However, 3D-printed food could be considered equal to the other ingredients because rarely is a whole meal is printed, it is usually parts of it. The 3D printing of food was not common by the time the consumer survey was executed as no 3D-printed food, dish or food ingredient were marketed for Finnish consumers.

Second consumer survey questionnaire was assembled of the questions regarding the raw materials and questions regarding the participants as background variables. A consumer survey was distributed as an online survey with Surveypal (Surveypal Inc., Tampere, Finland). Consumers were given instructions to think about the raw material as an ingredient of a food product or in cooking. Only the name of the raw material was given in the form with no further information. Consumers’ willingness to try and opinion on usage and the conceptual properties of the raw materials was gauged with 7-point Likert scale. Prospects of the raw materials were investigated with statements such as “I could eat or cook made of this raw material” and “I would be interested in trying food made of this raw material”. The scale was verbally anchored from both ends (1 totally disagree, 7 totally agree). The pleasantness of the sensory characteristics of the raw materials was evaluated with 7-point hedonic scale from 1 (extremely unpleasant) to 7 (extremely pleasant). Participants evaluated pleasantness of appearance, odor and flavor, and feel in fingers and mouth. Each participant evaluated three randomly presented raw materials.

Consumers’ attitudes and values were collected as background variables. Attitudes towards new food was studied with the food neophobia scale (FNS) [30]. The FNS consists of 10 items with a seven-category response scale ranging from “disagree strongly” (1) to “agree strongly” (7). Half of the items are reversed, therefore scoring of these items was reversed before calculating the FNS score as a sum of all the item scores. The Finnish translation of the FNS was used with minor revisions in wording as published in a Finnish textbook [32,33]. Participants were divided into three groups based on their FNS score. The three groups were formed following the procedure by Knaapila et al. [34]. Participants with low FNS scores (10–24) were regarded as “food neophilics”—score 25–39 indicates “median group” and score 40–67 indicates “food neophobics”. Gender, level of education, part of Finland where the participant lives, type of neighborhood, type of the household and diet were collected with category scales. Participants were asked to inform on whether they grow vegetables, berries or fruits themselves or pick berries, mushrooms or other ingredients from nature for food or cooking (Yes, I grow/Yes, I pick/No, I do not grow or pick). The questionnaire was completed anonymously.

Volunteer participants responding to the survey were recruited by a commercial supplier of consumer surveys. A total of 1014 participants were recruited to obtain adequate amount of replies for each raw material. To ensure variation of the background of the participants, quotas for demographic variables were generated. Usually women tend to participate in the surveys more eagerly than men. In our study, a minimum of 30% male participants was pre-established to secure adequate representation of both genders. In addition, participants from different parts of Finland were recruited from southern, western, eastern and northern parts of Finland at approximately 25% each. Highly educated participants are often overrepresented in the sample. Therefore, a quota for a minimum of 50% lower educated participants was created. Inclusion criteria were interest to participate and responsibility of groceries of the household alone or together with others. The group of participants was not representative of the Finnish population. The commercial supplier rewarded the participants according to their normal procedure.

### 2.4. Statistical Analysis

Results of consumer survey 2 were statistically analyzed using SPSS (IBM SPSS Statistics, 26, IBM Corporation, Armonk, NY, USA). Comparisons between the distribution of the results were performed to analyze differences between samples and respondent groups. Independent samples *t*-test or one-way analysis of variance (ANOVA) with Tukey’s or Tamhane’s post hoc test were used for variables and groups with normal distribution of categories. Most of the distributions were not normal. Therefore, a Mann–Whitney U-test and Kruskal–Wallis 1-way ANOVA methods with pairwise comparison were applied. The pairwise comparison was performed and significance values were adjusted by the Bonferroni correction for multiple tests. The criterion for statistical significance in all tests was *p* < 0.05.

## 3. Results

Participants of both consumer surveys were volunteer Finnish consumers. Demographic information of the participants is presented in Table 1. A more detailed description of the participants is presented in Section 3.1. and Section 3.2.1.

### 3.1. Consumer Survey 1

Participants in consumer survey 1 were not predetermined with quotas. A total of 380 replies of volunteer participants was received. In total, 82.8% of the participants were women, 15.4% men and 1.8% other or did not want to specify gender. Consumers who participated were 18 to 81 years old and mean age was 42.9 years. Detailed information of the participants is presented in Table 1.

Plant-based ingredients were the most interesting according to consumers similarly to the type of suggestions. Only 24 of the 81 raw materials were of animal or insect origin and only the 24th of the raw materials in order of the most interesting ones was of animal origin.

Although nettle is well known in Finland [35], it is not commonly used in current cuisine. There is potential for future usage, since 57% of the respondents were interested in using nettle. It was the most interesting of the wild vegetables. Berry bush leaves were almost as preferred as nettle (56%). Black currant leaves are to some extent used in seasoning in certain traditional food and drinks, but the amounts consumed are very small. Pine and spruce shoots are traditionally used as medicine and are, for example, eaten to avoid C-vitamin deficiency. In recent years, small food companies have started to produce food products from spruce shoots, but these are not widely used. Usage of pine or spruce shoots in cooking at homes is very rare. The fourth interesting wild vegetable was clover, which is used in salads, soups or herbal drinks, but the usage is marginal [36].

The following group after wild vegetables was the wasted parts of vegetables. The most preferred raw material representing this group was leaves and stem of broccoli and cauliflower (49%). These are used to some extent together with the other parts of broccoli or cauliflower, but they compose a great amount of wasted food material especially during the domestic season when the prices are lower. Root vegetable tops (41%), fruit seeds (35%) and potato peels (32%) represent the same group of raw materials and were also quite popular.

The most preferred raw materials of animal origin were whey protein (32%) and milk protein (30%). These are already commonly used in food products by manufacturers but not generally used by individuals at home. Synthetic meat (18%) was the first raw material of animal origin after proteins mentioned earlier. In this context it was considered to be synthesized animal cells. Eggshells were the following animal-based raw material and 15% of the respondents regarded them interesting.

Insects were not preferred by the participants. At the time of the survey, insects were not allowed to be sold as food in Finland, but it was decided that legislation would change from the beginning of 2018. Therefore, a lot of news and discussion about insects in the food sector has been underway. Thus, the interest towards insects might have been higher. Crickets were the most interesting insects in the survey and 26% of the respondents were interested in using them in food products and cooking. Beeswax was interesting to 24% of the respondents. Beeswax is used as a food additive—e.g., in coating certain fruits. Ants were the third interesting of the insect-based raw materials but only 12% of the respondents choose it as an interesting one. Ant eggs and mealworms were similarly interesting (12%). Since the beginning of 2018, mealworms have been allowed to be sold as food in Finland but are not widely used.

3D-printed food was equated with raw materials since with this method parts of dishes can be produced. In the printing process either plant- or animal-based ingredients can be utilized. Otherwise wasted materials could, for example, be printed to accepted food products. However, 3D-printed food was not considered interesting by the respondents of the study. It was considered equally interesting as a synthetic meat. This result, together with the top of the list including many wild vegetables and herbs, indicates naturalness as an important factor for consumers when considering the new interesting raw materials. The same findings indicate familiarity or tradition to be another significant element when choosing new ingredients as food which is also shown in previous research [37]. The top raw materials are traditionally used as medicine or in cooking. They are also commonly found plants on Finns’ own yards. Preferring plants over animal-based raw materials might signify either the importance of ecological aspects or unfamiliarity regarding edible insects. Insects were under discussion at the time of the survey and therefore were hypothesized to be trendy. Healthiness together with tastiness are considered when choosing raw materials for food [38]. These above-mentioned factors were included in consumer survey 2 to further investigate consumers’ opinion on the subject.

Results of consumer survey 1 were used as screener of the raw materials for the consumer survey 2 as described in Section 2.2.

### 3.2. Consumer Survey 2

#### 3.2.1. Participants of Consumer Survey 2

Detailed information of the participants in consumer survey 2 is presented in Table 1. In total, 58.3% of the participants were women, 41.5% men and 0.2% other or did not want to specify gender. Consumers who participated were 18 to 80 years old and mean age was 50.2 years. The majority (87.3%) of the participants had higher than basic education. There was an even distribution of respondents from different parts of Finland. Representatives of city life and countryside were featured. People from households of only adults formed the majority of the respondents; 22.6% of the participants had children in their household. Most (80.5%) of the respondents had a mixed diet. Special diets for different allergies, intolerances or disease or weight control were mentioned as “other”. Participants were divided into three groups based on their FNS score as explained in methods (see Section 2.2): 51.3% formed the median group, 28.7% were defined as food neophobics and 20.0% as food neophilics. Picking ingredients from nature or participants growing them themselves was assumed to affect opinions regarding parts of the raw materials in question. Nettle, berry bush leaves, pine or spruce shoots and clover can be picked from nature in Finland and they are available around the country. They also grow in gardens. Furthermore, apples, berries and root vegetables among other food ingredients are grown in gardens. Growing raw materials themselves was assumed to make them more interesting and otherwise affect opinions regarding the conceptual characteristics and pleasantness. Picking berries, mushrooms or other ingredients from nature is quite common among participants; 63.7% reported picking ingredients from nature. Frequency of picking was not predefined. Growing vegetables, berries or fruits was not as common as picking ingredients. Only 28.6% reported growing food raw materials themselves. Amounts grown or area used for growing was not predefined.

#### 3.2.2. Consumers’ Opinion on Willingness to Try and Conceptual Characteristics

As the 1014 participants answered the questions regarding three randomly selected raw materials, there were 226–415 responses for each. Distribution of the responses of each statement regarding the possibility to use, willingness to try and the conceptual characteristics are presented in Figure 2A–H. Raw materials are presented in order of the interest according to consumer survey 1 so that the first five are plant-based and next five animal- or insect-based to get a view of the raw material groups based on their origin. According to consumer survey 1, the origin of the raw material (animal/insect or plant) was a significant factor for the respondents, thus it is relevant to examine these groups. Significant differences in the distribution of the responses between raw materials are presented with lower-case letters. Differences are reported with significance level *p* < 0.05.

Based on the results of consumer survey 1, differences in consumers’ opinions between plant-based and other ingredients were assumed. This hypothesis was not thoroughly verified by the results of consumer survey 2. Beeswax deviated from other raw materials of insect or animal origin. Beeswax is currently used in food as a coating agent for certain foods. However, it is assumed that the majority of average consumers are not aware of this. Distribution of agreement on the statement “I could eat or cook food made of this raw material” was the same with crickets, synthetic meat, eggshells, ants and 3D-printed food. The majority of the respondents at least somewhat disagreed with the statement regarding these ingredients (Figure 2A). Consumers’ opinions on nettle and berry bush leaves were the opposite. The majority of consumers at least somewhat agreed that they could eat or cook food made with nettle (70%) or berry bush leaves (71%). Consumers are responsive to nettle, since 42% of the respondents totally agree they could eat or cook food from that. Finnish consumers were not ready to adopt insects into their everyday diet. Only 23% of the respondents to some extent agreed that they could eat crickets and 43% totally disagreed. Similar responses were given for ants—only 17% agreed to some extent and 47% totally disagreed. There was a distinct difference to other raw materials of insect or animal origin in disagreement with the statement. Regarding beeswax, 14% of consumers totally disagreed whereas, regarding ants, crickets, eggshells and synthetic meat, 47%, 43%, 36% and 27% totally disagreed, respectively—i.e., they would not eat the raw material in question. Finnish consumers are not ready to adopt 3D printing as a food manufacturing practice. The majority (62%) of respondents at least somewhat disagreed that they could eat 3D-printed food. They were not even willing to try 3D-printed food. Over a third (36%) of the respondents totally disagreed—i.e., they would not be willing to try 3D-printed food (Figure 2B).

Willingness to try (Figure 2B) shows similar differences for the non-plant-based raw materials, as assessed via the statement “I could eat or cook food…” (Figure 2A). Consumers are not willing to try crickets, ants, eggshells, synthetic meat or 3D-printed food. Slight differences in the opinions on the plant-based raw materials were discovered compared to the statement regarding whether they could eat or cook those materials. Nettle and berry bush leaves were considered as the most credible (Figure 2C). For nettle, 73% of the respondents and 69% for berry bush leaves stated at least somewhat agreed to their credibility. These raw materials are already marginally used for food, which might explain the higher credibility. Insects are not seen as credible for usage as food. The allowance of crickets to be sold as food might explain the slightly, though not significantly, higher credibility compared to ants. However, the majority of the respondents consider insects as not credible food; 54% of respondents at least somewhat disagreed that crickets are credible and 66% that ants are credible. Distribution of replies regarding beeswax and credibility is similar to “could eat” and willingness to try. Finnish consumers do not consider 3D printing as a credible technology for food preparation; 3D-printed food was regarded as least credible together with synthetic meat, eggshells and ants. Only 11% of the respondents agreed to some extent that 3D-printed food is credible.

Consumers’ opinion on the nutritional value of the raw materials was investigated. Any information about the nutrient content of the raw materials was not given in the questionnaire. Distinction between plant-based and other raw materials was not conspicuous. Nettle was considered as most nutrient-rich (81% at least somewhat agree), significantly different from all others (Figure 2D). According to consumers’ opinion, clover was less nutrient-rich compared to nettle and berry bush leaves and beeswax and crickets were considered as nutrient-rich as clover. Consumers did not regard 3D-printed as nutritious food. Synthetic meat and 3D-printed food were considered comparable and the least nutritious compared to the raw materials which are not produced but are derived by growing or as side streams of food preparation.

Nettle and berry bush leaves were also regarded as the most ecological raw materials (Figure 2E). The raw material representing side streams, broccoli and cauliflower stems and leaves and root vegetable tops were regarded as equally ecological compared to wild vegetables apart from nettle. Plant-based raw materials were highly regarded as ecological and more ecological than others; 79–89% of respondents at least somewhat agreed that the plant-based raw materials are ecological. Insects together with beeswax and eggshells formed the next ecological group; 53–63% of respondents at least somewhat agreed they are ecological. Synthetic meat was less regarded as ecological than the two previous groups but more than 3D-printed food. The raw materials which can be picked from the nature were also considered as natural and plant-based raw materials above others (Figure 2F). Only 5–7% of the consumers disagreed to some extent that the plant-based raw materials are natural. Insects were also regarded as natural but significantly less so than the plant-based raw materials. Over half (55–56%) of the respondents regarded insects as natural. As assumed in the wording, synthetic meat was not regarded as natural by consumers. In total, 75% of the respondents disagreed with the statement. Additionally, 3D-printed food was not regarded as natural, as only 82% of the respondents disagreed to some extent with the statement.

Based on the public discussion, it was assumed that insects could be considered trendy. Crickets were one of the first insects approved [11]. Half (51%) of the respondents at least somewhat agreed that crickets are trendy (Figure 2G). However, 23% of the respondents totally disagreed with the idea that crickets are trendy. Wild vegetables, except clover, were considered the most trendy. Eggshells, which are part of Finns’ everyday cooking, were considered the least trendy but this was not significantly different from synthetic meat or 3D-printed food.

Nettle was regarded as the most nutrient-rich (Figure 2D) and was also one of the raw materials regarded as the most healthy (Figure 2H). Similarly, synthetic meat and 3D-printed food were considered the least healthy. Beeswax is used as a coating agent and has been reported not to interact with human digestion at all [39]. However, 50% of the respondents at least somewhat agreed that beeswax is healthy. Synthetic meat and 3D-printed food were considered as the least nutrient-rich and also the least healthy; only 10% and 7%, respectively, to some extent agreed that they are healthy.

#### 3.2.3. Consumers’ Image of Sensory Properties

Participants evaluated the pleasantness of the raw materials without any additional information given in the question. The responses to the questionnaire were based on either a recollection of the raw material if the person had previous experience of it or an image if the respondent had no experience of the raw material. Appearances of all the plant-based raw materials were evaluated as more pleasant compared to the raw materials of insect or animal origin apart from beeswax. Additionally, 3D-printed food was seen as less pleasant than plant-based raw materials and beeswax. The appearance of clover and berry bush leaves was the most pleasant (Figure 3A). Most (82%) of the respondents regarded the appearance of clover as at least somewhat pleasant. A proportion (71%) of respondents regarded the appearance of berry bush leaves as at least somewhat pleasant. Pine or spruce shoots, nettle, broccoli and cauliflower leaves and stems and root vegetable tops were regarded as pleasant—over 50% of the respondents evaluated these as at least somewhat pleasant. The appearance of beeswax was evaluated as not pleasant nor unpleasant, but the difference to nettle or broccoli and cauliflower leaves and stems and root vegetable tops was not significant. Eggshells were evaluated as slightly unpleasant; 47% evaluated t as somewhat unpleasant. Synthetic meat and 3D-printed food were also regarded as slightly unpleasant and were not significantly different from eggshells. The 3D printing of food has, to date, been uncommon and it was assumed that most of the consumers had no experience of 3D-printed food. Nevertheless, it was assumed that consumers would have thought food can be printed as any kind of form and therefore the appearance was evaluated as pleasant. This assumption was as discovered false. The appearance of ants was perceived as at least somewhat pleasant by 79% and extremely unpleasant by 40% of the respondents and crickets by 73% and 46%, respectively.

Pleasantness of the aroma and flavor (Figure 3B) of the raw materials of different origins deviated similarly as related to the pleasantness of appearance. Beeswax was at the same level as plant-based raw materials in terms of pleasantness of aroma and flavor. Three-dimensional-printed food was evaluated as less pleasant than plant-based raw materials and beeswax. Aroma and flavor of berry bush leaves were evaluated as somewhat pleasant (50%) or as extremely pleasant (27%). Insects, synthetic meat, eggshells and 3D-printed food were the least pleasant. The proportion of responses of the unpleasant categories was significantly larger. Respondents who evaluated the aroma and flavor of insects as pleasant were a small minority; 16% indicated some degree of pleasantness to the aroma and flavor of crickets and only 9% for ants. Over a third (39%) of the respondents evaluated the aroma and flavor of crickets as extremely unpleasant and 43% did so for ants. Participants were not quite as critical about synthetic meat and eggshells. Distribution of these two raw materials was very similar. Pleasantness of aroma and flavor was not significantly different from crickets but was more pleasant compared to ants. Synthetic meat and eggshells were not regarded as having pleasant aromas and flavors. One-fourth of respondents evaluated the pleasantness of aroma and flavor as extremely unpleasant. Only 18% of respondents indicated some level of pleasantness to synthetic meat and 19% to eggshells. Finnish consumers are not familiar with 3D-printed food and opinions regarding this raw material are not as strong. One-third of the participants evaluated the aroma and flavor of 3D-printed food as not pleasant nor unpleasant. However, it was one of the most unpleasant raw materials in the study. One-fourth (26%) of respondents regarded the aroma and flavor of 3D-printed food as extremely unpleasant. 

Participants evaluated how pleasant the feel of the raw material in fingers and mouth is (Figure 3C). There was no information about the preparation of the raw material, but the questionnaire regarded raw materials in food and cooking. Thus, participants could imagine the raw material in question either as raw or prepared in some way. In relation to nettle, some of the respondents might have imagined the plant as raw and for that reason regarded it as very unpleasant. In total, 43% of the respondents regarded the feel of nettle as at least somewhat unpleasant. Berry bush leaves, clover and broccoli and cauliflower leaves and stems and root vegetable tops were the most pleasant raw materials in terms of feeling. Berry bush leaves were evaluated as pleasant by 63% of the respondents, clover by 59% of respondents and broccoli and cauliflower leaves and stems and root vegetable tops by 56% of the respondents. The pleasantness of the feel of beeswax was at the same level with nettle, pine or spruce shoots and broccoli and cauliflower leaves and stems and root vegetable tops. The pleasantness of the feel of raw materials of insect or animal origin together with 3D-printed food were evaluated as less pleasant compared to others. The feel of synthetic meat, eggshells and 3D-printed was equally pleasant. The majority of the respondents evaluated these raw materials as unpleasant; 58% regarded 3D-printed food as unpleasant, 60% did so for synthetic meat and 66% for eggshells. According to Finnish consumers, ants feel the most unpleasant of the investigated raw materials together with crickets. Ants feel unpleasant according to 77% of respondents and crickets do to 81% of respondents. 

#### 3.2.4. Differences between Opinions’ of Consumer Groups

Differences in responses between consumer groups were examined. Consumer groups with different demographic backgrounds were compared. Gender, education level, part of Finland living in, type of neighborhood living in, type of household and diet were used for grouping. Furthermore, differences between groups formed by food neophobia scores were investigated. It was assumed that picking and growing food ingredients oneself could affect opinions regarding investigated raw materials. Therefore, this background information of the respondents was also used to form consumer groups for comparison. The number of the respondents who specified gender as other or did not want to specify gender was small and this group was not compared as a group of gender. The group of other diets was small and heterogenic including diets from different reasons (i.e., weight control and allergies), thus respondents who indicated diet as other were not compared. Number of lacto-ovo-vegetarian and vegans among respondents was low, therefore these groups were not included in the comparison of diets. Results of comparison of respondent groups are presented in Appendix A
Table A1, Table A2, Table A3, Table A4, Table A5, Table A6, Table A7, Table A8, Table A9, Table A10 and Table A11. There were only few significant differences between consumers living in different parts of Finland or representing different types of households. Therefore, results of these groups are not presented in tables, but are explained in writing.

Women were more interested in trying nettle as a food and they were also more willing to eat or cook food with it. They also regarded nettle as more credible, nutrient-rich, ecological, trendy and healthy. There were also differences between age groups regarding attitude towards nettle. Respondents of age 50–64 more strongly, compared to 18–34 and 35–49-year-old groups, agreed they could eat nettle. Younger adults (18–34 years old) considered nettle as less nutrient-rich, healthy and ecological compared to the 50–64-year-old group. They also evaluated the appearance of nettle as less pleasant compared to the 50–64-year-old group. Nettle was most credible to the 50–64-year-old group. Participants with a higher education level were more willing to eat or cook food nettle. Respondents living in rural areas evaluated the aroma and flavor of nettle as more pleasant compared to others. Respondents having plant-oriented mixed diet were more willing to try and eat nettle and evaluated it as more pleasant regarding sensory properties compared to respondents with regular mixed diet. They regarded nettle as more credible, nutrient-rich, trendy and healthy. A food-related closer connection to nature, i.e., growing food oneself, using raw materials or picking them from nature, has an impact on opinions regarding this type of raw material. Respondents who pick food ingredients were more willing to try and eat nettle and consider it more pleasant compared to those who do not pick it. They also regarded nettle as more credible, nutrient-rich and healthy. Food neophobics were less willing to try and eat nettle and consider it less credible, nutrient-rich, ecological, natural, trendy and healthy. Furthermore, food neophobics evaluated nettle as less pleasant regarding appearance, aroma and flavor, compared to food neophilics and the median group, and feel in fingers and mouth less pleasant compared to food neophilics. Comparison of different consumer groups’ opinion on nettle are presented in Appendix A
Table A1.

Women were more willing to try and eat berry bush leaves and consider them more credible, nutrient-rich, ecological and trendy (Appendix A
Table A2). Moreover, female respondents evaluated the appearance, aroma, flavor and feel of berry bush leaves as more pleasant. The youngest group was significantly different from the 50–64-year-old group regarding whether they could eat, or were interested in trying berry bush leaves, considering whether they are credible or trendy and the pleasantness of feel, whereas both younger age groups were different from the 50–64-year-old group regarding nutrient richness and pleasantness of aroma and flavor. Respondents living in the western part of Finland were not as willing to try berry bush leaves in food as respondents from other parts of Finland (Kruskal–Wallis H = 16.469, *p* = 0.001 with mean ranks of 107.99 for West, 143.50 for South, 159.01 for East and 148.92 for North). Respondents from rural areas evaluated the aroma and flavor and feel of berry bush leaves as more pleasant compared to respondents from the center of large cities. The only difference between respondents from different types of household was in naturalness; respondents from adult households regarded berry bush leaves as more natural compared to single households (Kruskal–Wallis H = 7.633, *p* = 0.022 with mean ranks of 121.70 for single households, 149.16 for adult households and 145.50 for families with children). Respondents who grow food ingredients themselves considered berry bush leaves as more ecological and natural. Furthermore, consumers who pick food ingredients from nature were more willing to try and could eat berry bush leaves, considering them more credible, ecological, natural and trendy, and evaluating them as more pleasant. Food neophilics evaluated the sensory characteristics of berry bush leaves as the most pleasant and healthy. All the FNS groups were different in relation to the statement “I could eat or cook food made of…”, “I consider this credible”, “This raw material is natural”. Food neophilics most strongly agreed and neophobics least strongly agreed with these statements. Food neophobics were less interested to try berry bush leaves as food and consider them less ecological compared to the other FNS groups. Food neophilics regarded berry bush leaves as more nutrient-rich and trendy compared to food neophobics.

Female respondents were more willing to try pine or spruce shoots and evaluate them as more pleasant in appearance (Appendix A
Table A3). Furthermore, they regarded this raw material as more credible, nutrient-rich, natural, trendy, healthy and pleasant. The 50–64-year-old respondents, compared to 35–49-year-old respondents, were more willing to try pine or spruce shoots and also more strongly agree with the idea of eating or cooking food made of it. They also considered pine or spruce shoots as more nutrient-rich and ecological compared to others. Younger adults do not consider pine or spruce shoots as healthy as 50–64-year-old people. Representatives of adult households regarded pine or spruce shoots as more ecological compared to representatives of single households (Kruskal–Wallis H = 9.313, *p* = 0.010 with mean ranks of 98.97 for single, 128.43 for adult household and 108.32 for families with children). Consumers having plant-oriented mixed diet regarded pine or spruce shoots as more nutrient-rich and natural. Respondents who grow or pick ingredients for food themselves had more positive attitude towards pine or spruce shoots. Food neophilics were more willing to try and eat pine or spruce shoots, and consider them more credible, nutrient-rich, natural, trendy, healthy and pleasant compared to other FNS groups. Food neophilics and the median group regarded pine or spruce shoots as equally ecological but more than food neophobics. Food neophobics regarded pine or spruce shoots as less ecological than others. All FNS groups were different from each other in terms of whether they could eat or cook, willingness to try, credibility, trendiness and healthiness.

Women were more willing to try the leftover parts of vegetables as food raw material over men and also regarded them as more credible, nutrient-rich, ecological, natural and trendy (Appendix A
Table A4). Furthermore, pleasantness of aroma, flavor and feel were evaluated higher among women. The age group of 50–64 years old regarded the leftover parts of vegetables as more credible and nutrient-rich compared to the age group of 35–49 years old. Moreover, they evaluated the leftover parts as more pleasant regarding sensory properties and were more willing to try than the 35–49-year-old group. Respondents with a higher education evaluated the leftover parts of vegetables as looking and feeling more pleasant. Respondents living in the center of a larger city were more willing to try the leftover parts of vegetables and could eat and cook food made of them compared to respondents in rural areas. Furthermore, they regarded this raw material as more credible. Consumers from rural areas did not regard the leftover parts of vegetables as pleasant as others. Representatives of families with children considered this raw material as more ecological (Kruskal–Wallis H = 7.824, *p* = 0.020 with mean ranks of 103.78 for single households, 123.56 for adult households and 136.53 for families with children) and natural (Kruskal–Wallis H = 13.190, *p* = 0.001 with mean ranks of 100.88 for single households, 122.25 for adult households and 144.73 for families with children) compared to representatives of single households. Consumers from adult households evaluated the aroma and flavor (Kruskal–Wallis H = 9.991, *p* = 0.007 with mean ranks of 104.54 for single household, 133.95 for adult households and 109.70 for families with children) and feel (Kruskal–Wallis H = 7.137, *p* = 0.028 with mean ranks of 105.87 for single households, 131.49 for adult households and 113.46 for families with children) of this raw material as more pleasant compared to consumers living in single households. Consumers having a plant-oriented mixed diet were more willing to try and eat the leftover parts of vegetables. They considered this raw material as more credible, nutrient-rich and pleasant in aroma, flavor and feel. Respondents who grow or pick food ingredients from nature themselves evaluated the leftover parts of vegetables as more pleasant and nutrient-rich. There was a significant difference between pickers and non-pickers in willingness to try and eat, credibility, trendiness and healthiness. Leftover parts of vegetables were most pleasant and trendy to food neophilics. FNS groups differed similarly in willingness to try and credibility. Food neophobics had significantly lower agreement to statements “I could eat…”, “This is nutrient-rich”, “This is natural” and “This is healthy”.

There was a significant difference in opinions on clover between consumer groups based on gender in all the studied variables apart from nutrient-rich, trendy and healthy variables (Appendix A
Table A5). The youngest group was less willing to eat or cook food made of clover compared to 50–64-year-old consumers. Furthermore, the youngest group regarded it as least pleasant in sensory properties. The 35–49-year-old group regarded clover as the most nutrient-rich and natural. Respondents with higher education level indicated higher willingness to try and eat clover compared to basic education. Moreover, highly educated participants regarded clover as more ecological and natural. Clover is commonly growing in gardens in Finland. It is assumed that this plant is well-known by the consumers living in town houses. This might be one reason why consumers living in the center of a smaller city or municipality were more willing to try and eat clover in food compared to the consumers living in housing estates. There was a parallel difference between these two groups in credible, ecological, trendy and healthy variables. Consumers having plant-oriented mixed diets were more willing to try and eat clover and consider the raw material more pleasant. There was a difference between diets in all the investigated variables. Respondents picking or growing ingredients regarded clover as more nutrient-rich and more pleasant in appearance. The median group was more willing to try and eat clover compared to food neophobics and food neophilics more than median group and food neophobics. Furthermore, there was similar difference between FNS groups in ecological, natural and trendy variables. Food neophobics regarded clover as less credible, nutrient-rich and pleasant in appearance compared to others. Food neophilics evaluated the aroma, flavor and feel of clover as more pleasant compared to other FNS groups.

There was no difference between genders in willingness to try or whether they could eat crickets (Appendix A
Table A6). However, male respondents evaluated the appearance and feel of crickets as more pleasant compared to females. The youngest adults (age 18–34) were more willing to try and consider crickets as a possible part of their diet compared to 50–64-year-old participants. Respondents with a higher level of education regarded crickets as more credible and ecological compared to respondents with a basic education. Representatives with a higher or intermediate level of education regarded crickets as more nutrient-rich, natural, trendy and healthy. Crickets were evaluated as looking, smelling and tasting more pleasant by respondents with higher education levels. Consumers living in the center of large cities in Finland were more willing to try and adopt crickets in their diet compared to consumers living in a rural area. Furthermore, they considered them as more credible and nutrient-rich. Representatives of families with children could more likely eat crickets compared to single household representatives (Kruskal–Wallis H = 6.280, *p* = 0.043 with mean ranks of 107.86 for single households, 116.48 for adult households and 136.41 for families with children). Participants with plant-oriented mixed diets regarded crickets as more credible, nutrient-rich, ecological, trendy and healthy. They also evaluated the appearance of crickets as more pleasant. Consumers who pick food ingredients from nature regarded crickets as more nutrient-rich and healthy. Food neophilics were more willing to try and eat crickets compared to the median group and median group more than food neophobics. A similar difference was also in all the other variables except trendy where the median group was not different from food neophobics.

Beeswax was regarded as more ecological and natural by women (Appendix A
Table A7). The 50–64-year-old respondents evaluated beeswax as more appealing in all the sensory properties compared to the 35–49-year-old group. Pleasantness of feel of beeswax was evaluated higher also by the oldest age group (65–80 y) compared to 35–49-year-old participants. Representatives of families with children regarded beeswax as more natural compared to single (Kruskal–Wallis H = 7.295, *p* = 0.026 with mean ranks of 105.56 for single, 122.23 for adult household and 132.89 for families with children). Respondents having plant-oriented diet were more willing to try and eat beeswax. Moreover, they regarded it as more credible, nutrient-rich, natural, trendy and healthy. Pleasantness of aroma and flavor was higher according to respondents having plant-oriented diet. Respondents who grow or pick food ingredients themselves are more willing to try and eat beeswax. Furthermore, they consider beeswax as more nutrient-rich, trendy and healthy as well as more pleasant aroma and flavor. Moreover, respondents who pick food ingredients from nature regarded beeswax as more credible, ecological, natural and evaluated it as looking and feeling more appealing compared to those who do not pick. Food neophobics were not as willing to try and eat beeswax as median group and neophilics. They regarded beeswax less credible and natural as well as aroma, flavor and feel of beeswax as less pleasant compared to other food neophobia groups. Food neophobics evaluated beeswax as less nutrient-rich, ecological, trendy, healthy and appearance of beeswax less pleasant compared to food neophilics. 

Youngest (18–34 y) consumers were more interested in trying synthetic meat and more willing to adopt it as a part of their diet compared to the 35–49-year-old group (Appendix A
Table A8). Furthermore, the youngest consumers regarded it as more credible and the aroma and flavor as more pleasant compared to the 35–49-year-old group. Respondents with a higher education level regarded synthetic meat as more trendy compared to respondents with a basic education. Consumers living in southern part of Finland agree more with the statement “I could eat…” than consumers from the north (Kruskal–Wallis H = 10.251, *p* = 0.017 with mean ranks of 130.55 for South, 111.08 for West, 129.76 for East and 98.57 for North Finland). Respondents who stated their place of residence as the center of a large city are more willing to try synthetic meat and regarded it as more credible compared to respondents living in a housing estate or rural area. Consumers from large cities regarded synthetic meat as more ecological compared to representatives of rural areas. Respondents having plant-oriented diets regarded synthetic meat as more credible and natural. Food neophobics are less interested in trying synthetic meat compared to others. Furthermore, they regarded synthetic meat as less ecological, trendy, healthy and pleasant compared to others. Food neophilics regarded synthetic meat as more nutrient-rich compared to others and more credible compared to food neophobics. 

Consumers from East Finland could more potentially eat eggshells (Appendix A
Table A9) compared to consumers from the north and regarded eggshells as trendier (Kruskal–Wallis H = 9.564, *p* = 0.023 with mean ranks of 111.26 for South, 104.72 for West, 138.31 for East and 105.20 for North Finland) and feeling more pleasant (Kruskal–Wallis H = 15.828, *p* = 0.001 with mean ranks of 121.79 for South, 100.74 for West, 140.84 for East and 98.68 for North Finland) compared to consumers from the western and northern parts of Finland. Consumers who pick food ingredients from nature regarded eggshells as more natural, healthy and feeling more pleasant and they could more potentially eat eggshells as a part of their diet. Food neophilics are more willing to try and eat eggshells compared to others. Furthermore, they regarded eggshells as more nutrient-rich, ecological, trendy and healthy and looking, smelling and tasting less pleasant compared to other FNS groups. Food neophobics evaluated the pleasantness of feel of eggshells lower compared to food neophilics. Food neophobics regarded eggshells as less credible compared to others. Opinions on naturalness of eggshells were different between all the food neophobia groups.

Men were more willing to try and eat ants compared to women (Appendix A
Table A10). Men evaluated the appearance, aroma, flavor and feeling of ants as more pleasant. Furthermore, male respondents regarded ants as more credible, nutrient-rich, ecological, natural and healthy. Participants with higher education levels could more potentially eat or cook food made of ants, evaluated ants as more pleasant and regarded them as more nutrient-rich and trendy compared to participants with a basic education. Respondents with a basic education regarded ants as less credible, ecological, natural and healthy compared to other participants. Respondents living in South or East Finland consider ants trendier compared to respondents from the western part of Finland (*p* = 0.015, mean for West 3.25, East 3.55, South 3.64 and North 3.74). There were no significant differences between consumer groups based on age, place of residence, diet or growing or picking ingredients by oneself. Opinions of food neophobia groups regarding ants were significantly different in all the investigated variables. Food neophobics were less willing to try and eat ants and evaluated them as less pleasant compared to others. Food neophilics regarded ants as more credible, ecological, natural, trendy and healthy compared to others. Food neophilichs considered ants as the most nutrient-rich and food neophobics as the least nutrient-rich.

Men regarded 3D-printed food as more nutrient-rich, ecological and natural (Appendix A
Table A11). Furthermore, men evaluated the appearance, aroma and flavor of 3D-printed food as more pleasant. Unlike assumed, the oldest (65–80 y) consumers regarded 3D-printed food as more natural compared to the 35–49-year-old group. Respondents living in adult households regarded 3D-printed food as feeling more pleasant compared to respondents living with children (*p* = 0.018, mean for families with children 2.53, single households 2.76 and adult households 3.11). Contrary to other raw materials in the research opinions of food neophobia groups, these results were more similar. However, food neophilics were more willing to try and eat 3D-printed food and evaluated the appearance as more pleasant compared to food neophobics.

## 4. Discussion

In this study we explored the attitudes of Finnish consumers towards possible ingredients for future food. Differences in opinions between female and male respondents were noteworthy. Females were more open to plant-based raw materials and also regarded the conceptual characteristic higher. In accordance with previous research, men were more interested in trying ants and perceived crickets as more pleasant compared to women [40].

The group of 50–64-year-old respondents was more open to plant-based raw materials compared to the youngest group. Differences between education levels were not as comprehensive. A higher level of education indicated more openness to nettle, clover, leftover parts of vegetables and insects. Growing food ingredients by oneself or picking ingredients from nature for food and cooking indicates a close relation to nature and close relation to food ingredients and their origin. This might be a reason for more open attitude towards the raw materials that can be found from nature—i.e., wild food plants and insects. All the raw materials are novel or presumably quite unfamiliar to most of the Finnish consumers since they are not widely used at present. Therefore, it was assumed that food neophobia would contribute to attitudes towards investigated raw materials. This was comprehensively correct for all the investigated raw materials. Furthermore, food neophobia also affects the conceptual characteristics, not only willingness to try, but also the potentiality to use in food and cooking or pleasantness.

Based on the findings of our study, Finnish consumers are open to using nettle and berry bush leaves as a part of their diet. These ingredients were also regarded as the most ecological, natural, trendy, healthy and nutrient-rich by the respondents. This might be explained by the tradition of the use of these ingredients, though the use at present is marginal [26]. Pleasantness or willingness to try and using wild greens have not to our knowledge been studied and these results give valuable insights on their usage as novel ingredients in the future. Reception of clover as food ingredient is not as positive as nettle and berry bush leaves. Nevertheless, 58% of consumers show some degree of interest to try this ingredient. Similarly, as a grass protein, clover might have potential as a protein source for novel foods [41,42]. Leftover parts of vegetables are also perceived as potential novel ingredients by Finnish consumers. Women are more willing to try leftover parts of vegetables. Previous studies have shown women express higher motivation towards avoiding and reducing food waste [43,44], which explains the gender effect.

Finnish consumers are very cautious about synthetic meat, as only 25% of the participants indicated some degree of willingness to try synthetic meat. This is in contrast with previous research on synthetic meat. Almost the same proportion of Belgian participants indicated strong interest in trying cultured meat [45]. Weinrich et al. [46] reported German consumers to be unenthusiastic to try cultured meat while 57% of the participants indicated interest to try. More than half (54%) of the Italian respondents were willing to try cultured meat [47]. However, Belgian, German and Italian respondents received, at the least, basic information about cultured meat before indicating their interest [45,46,47].

Similarly, Finnish consumers do not express high interest towards 3D-printed food, since 36% of the consumers totally disagree with the statement “I would be interested in trying food made of 3D-printed food”. This is in contrast with the results of Manstan and McSweeney [25], where consumers showed higher interest towards 3D-printed food over conventional. Akin to Finns, Swiss consumers have negative attitude towards 3D-printed food [16]. However, well-designed communication has been shown to have the potential to positively shape consumers’ attitudes towards 3D-printed food [16]. Finnish consumers did not consider 3D-printed food as healthy, whereas Canadians perceived 3D-printed food as healthier compared to conventional counterparts [25].

Crickets and ants were representatives of insects in this study. Finns are not willing to adopt insects into their diet. Almost half of the respondents disagreed with the idea trying either crickets or ants. This finding is in line with previous research of the attitudes of Western citizens towards insects [17,40,48,49].

The origins of the novel ingredients included in the study varied very much. Some ingredients, e.g., wild food plants, are traditionally used and thus might be more familiar to some consumers whereas other ingredients, such as insects or 3D-printed food, might be very unfamiliar to most. Information of the familiarity or prior knowledge and experience of the ingredients would have given valuable insights for the interpretation of the results, since food exposure and familiarity is shown to affect liking and consumption of food products [37]. Previous research indicated that information changes the attitude towards unfamiliar food [50].

Information of the place of residence was given but not any further specific information about the place where the participants live. This might have also given background to the familiarity of the ingredients of natural sources. Clover, nettle and berry bush leaves commonly grow in gardens and even people living in housing estates or in the city center of a smaller city might have them in their own backyard. This information might have supplemented the information about the familiarity of the ingredients.

Our results were obtained from participants’ image of potential novel food ingredients. The image is a pre-existing factor that food producers need to understand when considering which previously unfamiliar or lesser used ingredients to incorporate into new food products or meals. Furthermore, consumers’ beliefs and perceptions about certain conceptual characteristics of ingredients, such as naturalness or nutrient richness, might affect how they are accepted in differing product categories or brand positionings. Further research is needed to investigate the perceived pleasantness of the ingredients which could potentially be used in future food.

## 5. Conclusions

Our results showed that consumers differed in their opinions about possible ingredients of edible products and meal. Based on our research, females, 50–64 years old, and neophilic respondents were more open to plant-based materials than others. Study participants were cautious about synthetic meat, 3D-printed food and insects in their diet. The number of consumers with basic education was not equal to other levels, which was a limitation of the study. However, the educational qualifications are high among adults in Finland. Variation of the ingredients led to difficulty of presenting comparable pictures of the ingredients. Although appearance is a significant determinant in the opinion-formation process, we decided not to give any additional information apart from the name of the ingredient. However, participants had the opportunity to search information while completing the questionnaire since the questionnaire was completed online and this was not reported. In general, plant-based ingredients were more agreed to by the consumers regarding conceptual characteristics than the other ingredients. From this point of view, they may have potential for future food ingredients.

## Figures and Tables

**Figure 1 foods-09-01669-f001:**
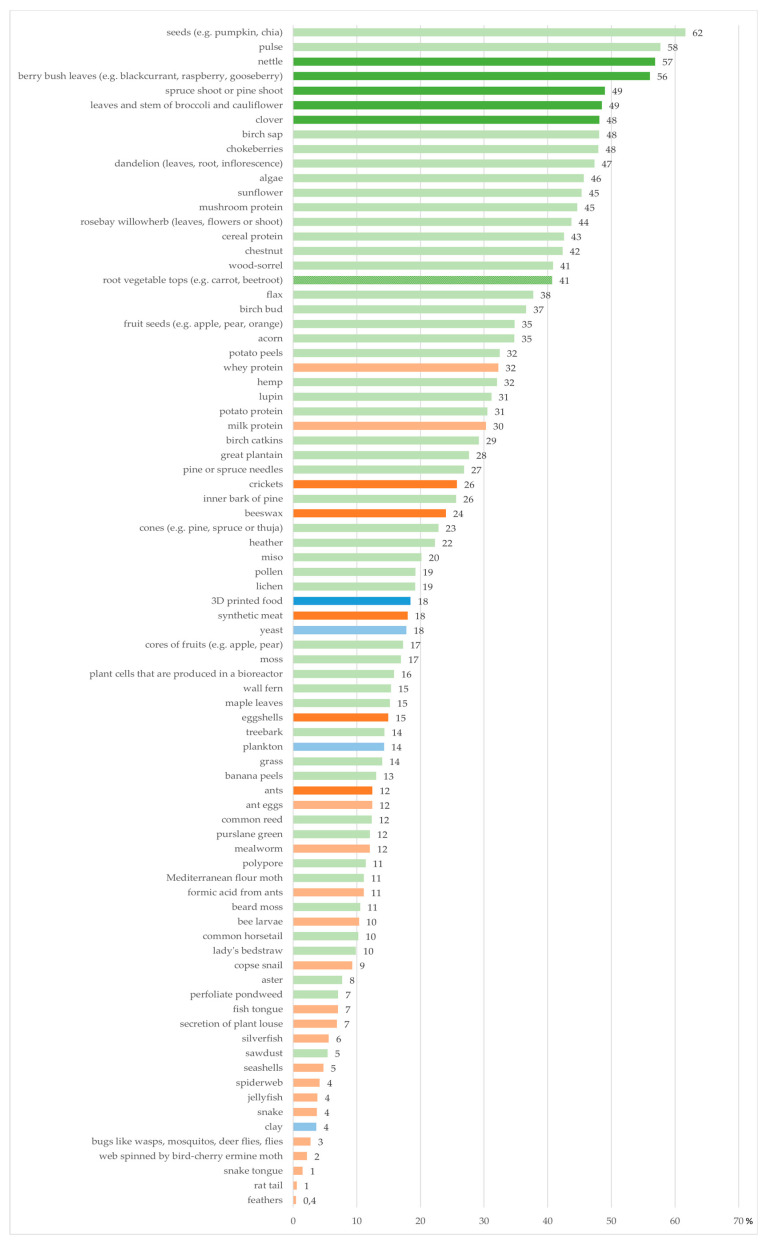
Raw materials listed in order of rated as most interesting by the participating consumers. Plant-based ingredients are marked with green, animal- and insect-based in orange and either or neither with blue. Darker shade indicates the raw materials chosen to consumer survey 2.

**Figure 2 foods-09-01669-f002:**
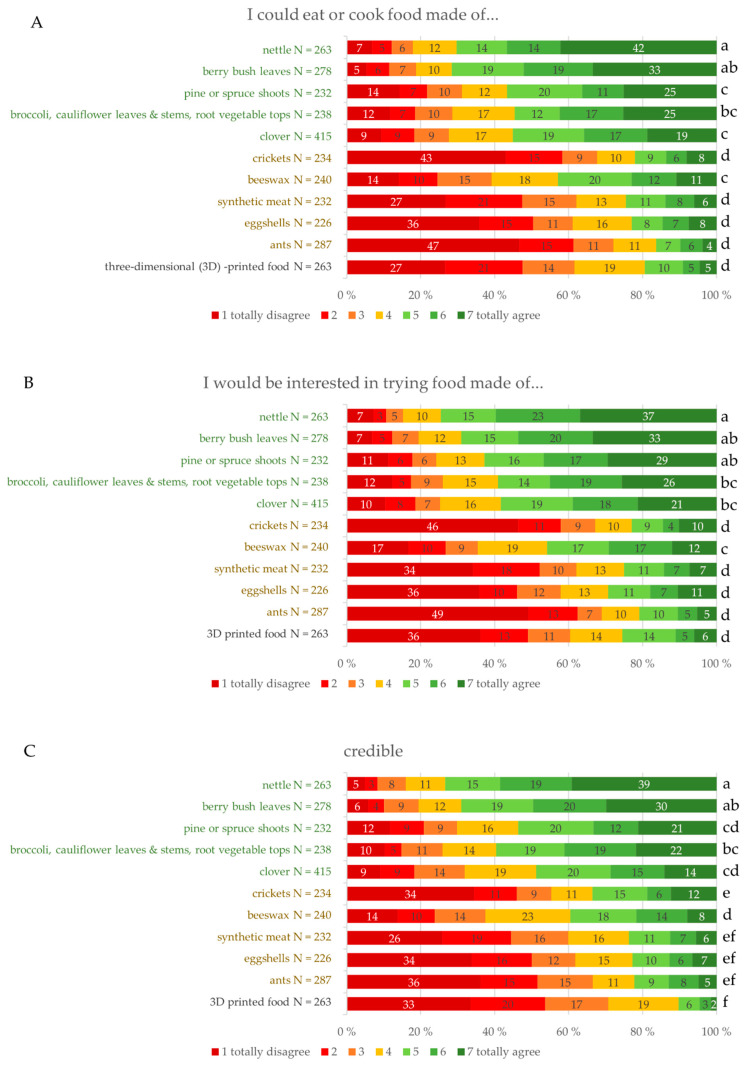
(**A**–**H**) Distribution (%) of the consumer opinion on possibility to use, interest to try and conceptual characteristics of the raw materials. Raw materials with significantly different distributions are marked with different lower-case letters. Significance level *p* < 0.05.

**Figure 3 foods-09-01669-f003:**
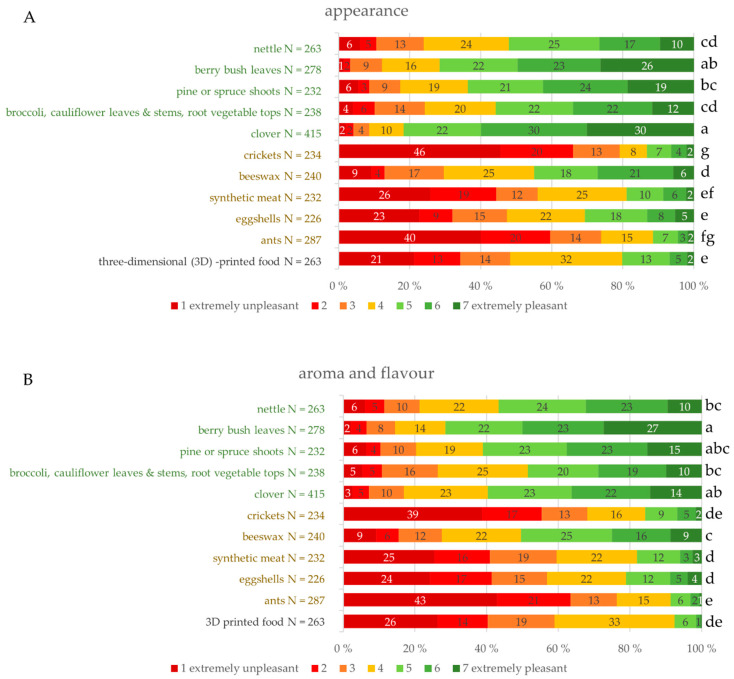
(**A**–**C**) Distribution (%) of the consumer opinion on pleasantness of appearance, aroma and flavor and feel in fingers and mouth. Raw materials with significantly different distributions are marked with different lower-case letters. Significance level *p* < 0.05.

**Table 1 foods-09-01669-t001:** Participants of consumer survey 1 and 2.

Consumer Survey 1 *n* = 380		Consumer Survey 2 *n* = 1014	
	% of the Participants		% of the Participants
Gender		Gender	
Female	83.4	Female	58.3
Male	15.5	Male	41.5
Did not want to specify gender	1.1	Did not want to specify gender	0.2
Age		Age	
Mean	42.9	Mean	50.2
18–34	35.5	18–34	21.9
35–49	29.7	35–49	24.9
50–64	25.5	50–64	30.7
65–80	9.2	65–80	22.6
		Education	
		Basic	12.7
		Intermediate	42.6
		higher level	44.7
		Part of Finland where lives	
		South	23.3
		West	24.2
		East	24.2
		North	28.4
		Neighborhood	
		center of a large city (over 100,000 inhabitants)	21.1
		center of a smaller city or municipality	26.9
		housing estate	33.1
		rural area	18.8
		Type of household	
		Single	32.1
		adult household	45.4
		family with children	22.6
		Diet	
		mixed diet	80.5
		plant-oriented mixed diet	11.1
		lacto-ovo-vegetarian	3.7
		Vegan	1.3
		Other	3.4
		Food neophobia group (Food Neophobia score)	
		food neophilics (10–24)	20.0
		median group (25–39)	51.3
		food neophobics (40–70)	28.7

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
