# Peer review of "Sensory and Conceptual Aspects of Ingredients of Sustainable Sources—Finnish Consumers’ Opinion"

_foods, 2020, doi:10.3390/foods9111669_

Round 1

Reviewer 1 Report

The work was well designed, described and discussed. 

In Materials and Methods, I would advise the authors to separate the phase in which consumers are asked for possible alternative ingredients, as a preliminary stage. And then talk about surveys 1 and 2.

Little shortcomings has been detected in the manuscript such as:

  • Line 26: "a" must be changed to "A"; after point is capitalized.
  • For references in the text, the number must be after the name of the authors (e.g. see lines 52-53; 67-68; 70-71)
  • I think that in line 66, "fruit and vegetables" must be removed. 
  • In decimal numbers, the decimal part is separated with a "."and not a "," (see all the manuscript, tables). 
  • Figure 1 (lines 105-106) is named before Table 1 (line 197) and yet appears later.

I hope that these suggestions will be useful to improve the presentation and discussion of the obtained results

Author Response

FOODS 975039

REVIEWER 1

REV: The work was well designed, described and discussed.

RESPONSE: We thank you for this kind comment.

REV: In Materials and Methods, I would advise the authors to separate the phase in which consumers are asked for possible alternative ingredients, as a preliminary stage. And then talk about surveys 1 and 2.

RE: Thank you for your advise. This part is revised to the manuscript.

Little shortcomings has been detected in the manuscript such as:

REV: Line 26: "a" must be changed to "A"; after point is capitalized.

RE: Thank you. This is revised to the manuscript.

REV:  For references in the text, the number must be after the name of the authors (e.g. see lines 52-53; 67-68; 70-71)

RE: Thank you for pointing this. Revised now.

REV: I think that in line 66, "fruit and vegetables" must be removed.

RE: Yes, You are right. This has been removed.

REV: In decimal numbers, the decimal part is separated with a "."and not a "," (see all the manuscript, tables).

RE: You are right. Revised now.

REV: Figure 1 (lines 105-106) is named before Table 1 (line 197) and yet appears later.

RE: Thank you for pointing this. We also revised this part of the manuscript.

REV: I hope that these suggestions will be useful to improve the presentation and discussion of the obtained results

RE: Thank you for Your valuable comments.

Reviewer 2 Report

Not clear why title cover only novel ingredients, while some of tested products are not defined as novel and are used in Noth-east of Europe for centuries

Weak explanation of Study 1 and study 2 testing designs and connections why results of both studies presented in one manuscript

L 199-203 and L214-216 Same information

L315-316. Not clear  hypothesis

Not clear from manuscript difference between 'pleasantness' and 'acceptability'

It is recommended to rewrite sentences which are very complicated to understand, for exampleL469 'Like pleasantness of other sensations, pleasantness of feel of raw materials of insect or animal origin together with 3D printed food were evaluated as less pleasant compared to others'

It is recommended to reorganize tables by deleting not necessary information 

Author Response

FOODS 975039

REVIEWER 2

REV: Not clear why title cover only novel ingredients, while some of tested products are not defined as novel and are used in North-east of Europe for centuries

RE: Thank you for these comments. Some of those have been used also in North-Europe, although in very small scale and not in the industrial products. But based on your comment we took the word novel away from the title.

REV: Weak explanation of Study 1 and study 2 testing designs and connections why results of both studies presented in one manuscript

RE: Study 1 and Study 2 were strongly connected and they form one solid story. Study 1 was used to select food items and ingredients to Study 2. We have now added the connection in details to the text.

REV:L 199-203 and L214-216 Same information

RE: We rewrote this part and removed the unnecessary repetition.

REV:L315-316. Not clear  hypothesis

RE: We understand that our text should have been sharper. Our hypothesis was that consumers differ in their opinions on plant-based and other ingredients. The key motivator for these surveys was to find out the potential barries and drivers for these food ingredients. This is now added also to the aim and also to results.

REV: Not clear from manuscript difference between 'pleasantness' and 'acceptability' 

RE: Pleasantness, referring to how much consumers like, was measured with 7-point hedonic scale that is used also to quantification of acceptability (Lawless and Heymann 2010). We understand that acceptability may be wider than hedonic based pleasantness. That was the reason why other conceptual characteristics such as healthy, trendy etc. where included to the questionnaire. Although consumers were not activated by food photos or real food items we activated them by written term.

REV: It is recommended to rewrite sentences which are very complicated to understand, for exampleL469 'Like pleasantness of other sensations, pleasantness of feel of raw materials of insect or animal origin together with 3D printed food were evaluated as less pleasant compared to others'

RE: Thank you for your advice. We have now rewritten the complicated sentences.

REV: It is recommended to reorganize tables by deleting not necessary information 

RE: Based on the comments from other reviewers we have removed tables from results to appendices.

Reviewer 3 Report

This work reports data from a very interesting research on consumer perception of different types of novel foods and the willingness to try new foods related to sustainability. In general, it is an interesting work, rich in information and supported by good statistical analysis. However, I found the paper too long and difficult to follow in reading; in fact, it is full of redundant parts that need to be eliminated. All the text has to be reduced, especially the results section. this latter part must be concise and must reported the main finding of the activity. I suggest moving the tables in the appendix and re-formulate the results considering the significant ones. The significance levels must be the typical 0.001, 0.01 and 0.05 and not only the 0.05 level. In the results some parts are discussed. The result section must contain only the research finding and not the literature references. Then, there is a discussion and the conclusions are missing. I would propose to combine the discussion with the results and include the conclusions. I have made some comments on the text: those relating to repetitions are to be applied to the whole manuscript. 
The work is good in terms of content, but needs to be improved in form and readability.

Please see the attachment for detailed comments.

Author Response

FOODS 975039

REVIEWER 3

REV: This work reports data from a very interesting research on consumer perception of different types of novel foods and the willingness to try new foods related to sustainability. In general, it is an interesting work, rich in information and supported by good statistical analysis. However, I found the paper too long and difficult to follow in reading; in fact, it is full of redundant parts that need to be eliminated.

RE: Thank you for these constructive suggestions. We have revised the manuscript further and moved also some parts to appendices.

REV: All the text has to be reduced, especially the results section. this latter part must be concise and must reported the main finding of the activity.

RE: Thank you for this constructive suggestion. We have revised the manuscript and reduced the text.

REV: I suggest moving the tables in the appendix and re-formulate the results considering the significant ones.

RE: We agree with the reviewer. We have moved the tables to appendix.

REV: The significance levels must be the typical 0.001, 0.01 and 0.05 and not only the 0.05 level.

RE: We agree with the reviewer. Significance levels include also 0.001 and 0.01, but the minimum level was 0.05. Based on the suggestion we have modified the manuscript.

REV: In the results some parts are discussed. The result section must contain only the research finding and not the literature references. Then, there is a discussion and the conclusions are missing. I would propose to combine the discussion with the results and include the conclusions.

RE: Thank you for this suggestion. However, we have used the template given by Foods in advance. We have now added Conclusions to the end of the manuscript.

REV: I have made some comments on the text: those relating to repetitions are to be applied to the whole manuscript.

RE: We have looked at those comments carefully and responded for each of them below.

REV: The work is good in terms of content, but needs to be improved in form and readability.

RE: Thank you. We have modified the manuscript.

From the pdf file

REV: ABSTRACT lines 16-18: In the abstract you can choose to describe both the first and second survey or to describe in a general way the results.

RE: We have modified the abstract based on your kind comment.

REV: Line 23: keywords: order alphabetically

RE: Thank you. We have modified the order of the key words.

REV: Line 33: Peano, Cristiana, Valentina Maria Merlino, Francesco Sottile, Danielle Borra, and Stefano Massaglia. "Sustainability for Food Consumers: Which Perception?." Sustainability 11, no. 21 (2019): 5955.

RE: Thank you for your suggestion. We have added this reference to the introduction.

REV: Line 64-66: Three dimensional food printing has suggested to have implications for future food development, including reducing food waste using food that is usually discarded, such as fruits and vegetables having poor quality in appearance fruits and vegetables [24]. Re-write.

RE: We have reformulated the sentence now.

REV: Line 67: rewrite

RE: We have rewritten the sentence.

REV: Line 68 and line 71: Date in brackets

RE:  Unfortunately we are not sure what this comment is pointing to.

REV: Line 82-84: When novel ingredients are introduced to consumers, it can potentially cause neophobia, that is fear and refusal of new food [29]. Neophobia limits individuals’ readiness to try new foods and thus restricts the marketability of new ingredients [30].

Line 85: tes, very intresting, but I I would try to conclude the previous part by explaining the common thread among all these novel foods and therefore talking about neophobia

RE: Thank you for this proposal. We have added this to the introduction.

REV: Line 93: explain in the aim also the context of the research

RE: Thank you for your suggestion. We explained the context of the research in the end of the introduction.

REV: Line 101: This was implemented in a local food fair autumn 2017 (total number of visitors approximately 20000).

RE: We have added more specific description in Preliminary stage 2.1

REV: Line 105-106: Join the 2 sentences.

RE: We have reformulated the text by joining the sentences.

REV: Line 111-117: writing this part better and in a more discursive way; short, disconnected sentences.

RE: We have modified the text to improve readability.

REV: Line 147: this last part must be summarised and reduced and written in a more concise manner

RE: More precise and modified

REV: Table 1: I think that a resume of the main socio-demographic characteristics of the 2 samples of consumers is needed

REV: Line 200-203: this is a repetition.

RE: Text was rewritten to remove repetition.

Line 204-207: already reported in the previous part.

RE: We have modified the text an removed the unnecessary repetition from the previous part.

REV: Table 1: change with dot apply in the whole document.

RE: We have applied this to whole document and changed as suggested.

REV: Table 1: NFS (10-24)

RE: The scale of FNS was 10-70.

REV: Line 215-216: Repetition

RE: We have modified the text to remove repetition.

REV: Lines 251-252: move this sentence when you talk about animal orogin raw materials

RE: Thank you for this suggestion which improves readability of the text. We have no revised the text according to your suggestion.

Lines 262-263: references are needed

RE: Thank you for your comment. References are added to support the findings of our study.

Lines 272-278: move sample description in the first paragraph of section 3

RE: Thank you for your comment. Since the samples were different in the two consumer surveys, they are described in different sections.

Line 315: (Figure 1.).

RE: We have corrected the writing error.

Line 391: (51 %)

RE: We have reformulated the text.

Table 6: Significance level is 0,05.

RE: We agree with the reviewer. Significance levels include also 0.001 and 0.01, but the minimum level was 0.05. Based on the suggestion we have modified the manuscript.

Line 707: (Table 10.).

RE: We have corrected the writing error.

Line 721: (Figure 2B.).

RE: We have corrected the writing error.

Line 722: (Table 11.).

RE: We have corrected the writing error.

Line 738: (Table 12.).

RE: We have corrected the writing error.

Line 760: Discussion

RE: We understood this refers to previous comment about reformulating discussion. We have now added Conclusions to the end of the manuscript.

Round 2

Reviewer 3 Report

I believe that the authors have worked well and made good changes to the text to make it clearer.

However, I remain of the opinion that the results section is too long and difficult to read. I propose either to shorten it again by including results+discussions in one section, or to divide it into paragraphs.

The conclusions lack the limits of the research.

I suggest fewer revisions to complete the work. 

Author Response

REV: I believe that the authors have worked well and made good changes to the text to make it clearer.

RE: Thank you for your kind comment.

REV: However, I remain of the opinion that the results section is too long and difficult to read. I propose either to shorten it again by including results+discussions in one section, or to divide it into paragraphs.

RE: Thank you for your suggestion. We have divided results into paragraphs and shortened the text.

REV: The conclusions lack the limits of the research.

RE: We have added limitations of the research to the conclusions. We recognise the challenge in the size of basic education group, which makes the comparison of the education levels more difficult. However, the share of basic educated adults is relatively lower than the share of higher education in Finnish population.